# The Impacts of Periconceptional Alcohol on Neonatal Ovaries and Subsequent Adult Fertility in the Rat

**DOI:** 10.3390/ijms25052471

**Published:** 2024-02-20

**Authors:** Sarah E. Steane, Danielle J. Burgess, Karen M. Moritz, Lisa K. Akison

**Affiliations:** 1School of Biomedical Sciences, The University of Queensland, St Lucia, QLD 4072, Australiad.burgess@uq.edu.au (D.J.B.); k.moritz@uq.edu.au (K.M.M.); 2Mater Research, The University of Queensland, South Brisbane, QLD 4101, Australia

**Keywords:** prenatal alcohol, ovarian reserve, estrous cycles, gene expression

## Abstract

Maternal exposures during pregnancy can impact the establishment of the ovarian reserve in offspring, the lifetime supply of germ cells that determine a woman’s reproductive lifespan. However, despite alcohol consumption being common in women of reproductive age, the impact of prenatal alcohol on ovarian development is rarely investigated. This study used an established rat model of periconceptional ethanol exposure (PCEtOH; 12.5% *v*/*v* ethanol) for 4 days prior to 4 days post-conception. Ovaries were collected from neonates (day 3 and day 10), and genes with protein products involved in regulating the ovarian reserve analyzed by qPCR. Adult offspring had estrous cycles monitored and breeding performance assessed. PCEtOH resulted in subtle changes in expression of genes regulating apoptosis at postnatal day (PN) 3, whilst those involved in regulating growth and recruitment of primordial follicles were dysregulated at PN10 in neonatal ovaries. Despite these gene expression changes, there were no significant impacts on breeding performance in adulthood, nor on F2-generation growth or survival. This contributes additional evidence to suggest that a moderate level of alcohol consumption exclusively around conception, when a woman is often unaware of her pregnancy, does not substantially impact the fertility of her female offspring.

## 1. Introduction

As women are increasingly delaying childbirth, and fertility is inversely proportional to age [1], it is important to determine environmental factors that may impact the establishment of the ovarian reserve. Ovarian reserve is defined as the number of oocytes stored within primordial follicles in the ovary, and determines a woman’s reproductive lifespan [2,3]. Unlike in males, these germ cells are finite and established prenatally in humans (~weeks 17–19 gestation) and neonatally in rodents (~postnatal day (PN) 3–4) [4,5]. They are gradually depleted throughout life, with a disproportionately high number undergoing apoptosis, while others are recruited into the pool of growing follicles [6]. For this pool of oocytes to be maintained and yet available requires a balance of factors to maintain their quiescence, as well as factors to initiate recruitment to each ‘wave’ of growing follicles (see Ford, et al. [7] for review).

There is evidence that the initial number of non-growing follicles established is proportional to the age at menopause, such that a high ovarian reserve predicts a later age at menopause [8]. There is wide individual variability in the number of primordial follicles established during this early developmental period, and many environmental factors may influence this number via the intrauterine environment [2,9]. However, the impact of prenatal alcohol exposure has been relatively overlooked [2,10]. This is a common maternal exposure in most Western countries, with many women consuming low to moderate amounts of alcohol around the periconceptional period and prior to pregnancy recognition [11,12,13]. Given this is difficult to study in a clinical population, animal models are required.

The current rat study examines the impact of a moderate dose of alcohol exclusively around the time of conception on subsequent ovarian development and fertility. We have previously published multiple negative impacts of this exposure on placental and fetal growth and development [14,15,16]. We have also shown more long-term, often sex-specific, effects in offspring derived from this model, including a diabetic-like phenotype [17], an increased tendency towards obesity [14,18], and impaired renal function [19]. This is despite the fact that only the precursor cells for the placenta and fetus, contained within the early embryo at the blastocyst stage, coincide with the actual timing of exposure. In the present study, we aimed to examine the impact of this periconceptional ethanol (PCEtOH) model on the expression of genes that produce factors regulating ovarian reserve in neonatal offspring, specifically those involved in the apoptotic pathway and primordial follicle activation. We also aimed to examine potential long-term impacts on estrous cyclicity and breeding performance in adulthood.

## 2. Results

### 2.1. Ovarian Gene Expression Changes in Neonates Exposed to Periconceptional Alcohol

Aside from differential expression of the tested endogenous control genes in PCEtOH versus control samples (as reported in the Materials and Methods section), genes of interest involved in the apoptotic pathway were dysregulated at PN3, but not at PN10 (Figure 1). Pro-apoptotic BCL2 associated X protein (*Bax*) had significantly increased expression in ovaries from offspring originating from PCEtOH-exposed pregnancies, while the pro-survival B cell leukemia/lymphoma 2 (*Bcl2*) gene showed a trend (*p* < 0.08) for increased expression. The ratio of the pro-survival BCL2 like 1 (*Bcl2l1*) to *Bax* was also significantly reduced in PCEtOH ovaries (Figure 1).

In contrast, genes regulating maintenance of the primordial follicle pool were dysregulated at PN10, but not at PN3 (Figure 2). Anti-mullerian hormone (*Amh*) and Inhibin alpha (*Inha*) were both increased in ovaries exposed to PCEtOH, suggesting a disproportionate number of larger follicles compared to controls.

### 2.2. Estrous Cyclicity and Breeding Performance in Adult Offspring

There was a subtle impact of PCEtOH exposure on estrous cyclicity in 3-month-old F1 offspring, with a subset from pregnancies exposed to alcohol having altered estrus patterns (Figure 3a). This resulted in a trend towards increased average cycle length and an average of one fewer cycles over the monitoring period compared to control animals (Figure 3b,c). There was no significant impact of PCEtOH exposure on F1 female breeding performance, measured by the number of pups born over two consecutive litters, or F2 pup weights and survival (Table 1).

## 3. Discussion

We measured expression of a subset of potential factors involved in regulating primordial follicle numbers to determine if these were affected by PCEtOH. Numbers of these non-growing follicles are controlled by a balance of factors that (a) regulate activation and recruitment to the pool of growing follicles or maintain quiescence via extrinsic (e.g., *Amh*, *Inha*) and intrinsic (e.g., *Pten*, *Cxcr4*, *Cxcl12*, *Stk11/Lkb1*) factors, or (b) regulate depletion via pro- and anti-apoptotic factors (e.g., *Bax*, *Bak1*, *Bcl2*, *Bcl2l1*, *Bcl2l11*, *Bbc3/Puma*). While many other factors are emerging as being important (see [20,21,22] for review), we chose this subset of relatively established regulators across the spectrum of possible primordial follicle fates (i.e., activation, quiescence or apoptosis).

We found subtle changes in the expression of genes of interest in neonatal ovaries from females originating from PCEtOH-exposed pregnancies, with those involved in the apoptotic pathway, and specifically pro-apoptotic, being upregulated at the earlier PN3 time-point, but not at PN10. Although this was only measured at the transcriptional level, and was not confirmed via measurement of Caspase-3 activity or localization, the lack of subsequent impacts on fertility suggest minimal impacts on folliculogenesis and reproductive capacity. Similarly, although increased levels of *Amh* and *Inha* occurred at the later PN10 time-point, suggesting a greater activation of follicles to the growing pool of follicles, and therefore a disproportionate number of preantral follicles compared to controls [23], the similar litter sizes produced by F1 females suggests that these changes were not sufficient to impact on fecundity.

We generally also saw very little evidence of impacts on estrous cyclicity in F1 female offspring, although a subset of females from PCEtOH-exposed pregnancies, but not control pregnancies showed increased cycle length (~2-fold increase) and, therefore, one less cycle on average over a 20-day period. However, there were no apparent impacts on subsequent breeding performance.

These results are similar to our earlier study using an episodic exposure of moderate alcohol at E13.5 and E14.5 of rat pregnancy, which also included assessment of follicle numbers in neonatal ovaries using histology and unbiased stereology [24]. Note that in this previous study, fetal ovaries were present at the time of exposure, whereas in this PCEtOH model, the fetal ovaries have not yet developed. However, we have previously shown that the maternal intra-uterine environment in the PCEtOH model is perturbed immediately following the exposure period, with both progesterone receptor (*Pgr*) and estrogen receptor alpha (*Esr1*) down-regulated during the period of receptivity in the rat (E5) and *Pgr* upregulated during the period of post-implantation decidualization (E7) [15]. Although circulating levels of progesterone and estrogen were unaffected over this peri-implantation period, progesterone-responsive decidualization genes were also dysregulated [15]. Therefore, despite fetal ovarian development occurring after the exposure treatment period, alterations in the maternal environment have the potential to alter developmental trajectories.

We did not directly quantify follicle numbers via histology in neonatal ovaries in the current study, including the non-growing primordial follicle pool. Therefore, the pro-apoptotic state of neonatal ovaries at PN3 and the putative enhanced activation of follicles at PN10 in the PCEtOH-exposed neonates could have reduced the initial ovarian reserve compared to controls. Given that modeling the age-related population of non-growing follicles in humans suggests a reduced initial ovarian reserve can predict an earlier age at menopause [8], this could suggest an earlier age of reproductive senescence in F1 females exposed to PCEtOH during pregnancy in this rodent model. We examined reproductive performance from ~3–6 months of age across two consecutive litters, a period of peak reproductive capacity, and found no differences between PCEtOH-exposed and controls. However, we did not monitor reproductive performance in F1 females as they aged. Sprague Dawley female rats typically start to become sub-fertile at ~8–10 months of age and undergo reproductive senescence at ~12 months of age [25]. Our systematic review [26] highlighted there were no studies exploring long-term fertility in either preclinical models or in women exposed to PAE. Therefore, future preclinical studies examining the impact of prenatal alcohol exposure on female offspring could include an aged cohort (i.e., >8 months of age) to examine the likelihood of premature reproductive senescence. In addition, females diagnosed with FASD in childhood could be monitored as they age to determine if the onset of menopause is affected.

Given that this model of PCEtOH has previously been demonstrated to result in sex-specific programming of long-term health, particularly in males, future studies could examine the impact of this moderate alcohol exposure on male reproductive parameters. This is especially relevant given the reduced nephron endowment reported in offspring from this model [19], and the fact that the kidney and testis both arise from the urogenital ridge during early fetal development [27]. There is some evidence a similar dose of alcohol during pregnancy can affect testis development and long-term reproductive performance in males [28,29,30], although to-date, no studies have examined exposure exclusively around conception.

To our knowledge, only one previous preclinical study in mice has examined the impact of prenatal alcohol exposure exclusively early in gestation (E5-11) on reproductive outcomes in offspring, although this was still after implantation [31]. The dose was comparable to our study (i.e., resulted in a blood alcohol concentration (BAC) of ~0.10–0.15%). However, they only examined the timing of vaginal opening as an indicator of puberty onset, reporting that this was delayed in female offspring from pregnancies exposed to ethanol compared to controls. Future studies could examine impacts of alcohol exposure exclusively during the periconceptional period on puberty onset in offspring, although given our results shown here on fertility, we suspect any impacts would be minor, if any.

An important finding from this study was that there was some evidence of regulation of candidate endogenous control genes by PCEtOH exposure. These should be stably expressed irrespective of treatment, and therefore suggests prenatal alcohol treatment can fundamentally alter the neonatal ovary at the molecular level. We recommend *Hprt* as a suitable, stably expressed endogenous control for future studies of gene expression in neonatal ovaries in prenatal alcohol studies.

Therefore, although previous studies using this model have reported other sex-specific impacts on F1 offspring metabolic and renal function, these results suggesting minimal impacts on female offspring reproductive function may provide some reassurance for women who have consumed a small amount of alcohol during their early pregnancy.

## 4. Materials and Methods

### 4.1. Animal Model and Tissue Collection

The housing conditions and treatment protocol for the rat model used in this study of alcohol exposure exclusively around conception have been previously described ([16,32]; Figure 4). Briefly, this involved a 12.5% *v*/*v* EtOH liquid diet provided ad libitum from 4 days prior to conception (embryonic day (E)-4)) to 4 days after conception (E4; PCEtOH; n = 11), with calorie-matched controls (Control; n = 11). All animals had a brief (1–2 day) pre-conditioning period to the diet prior to the addition of EtOH, resulting in high acceptance of the diet by all animals (as evidenced by minimal within-group variation and no significant between group difference in calorie intake and weight gain, as reported in [32]). At this dose, our aim was to produce a low-moderate exposure level, clinically relevant for most women. Peak BAC levels occurred 30 min after provision of fresh diet of 0.18 ± 0.04% on day E-2 and 0.25 ± 0.04% on E2 that rapidly declined to 0.07 ± 0.02% at 3h and 0.05 ± 0.02% at 5 h [32]. Although the peak BAC may appear high, this coincided with the initial period of maximal ingestion and quickly reduced to levels only marginally higher than levels found in occasional alcohol users [33]. Additionally, BAC levels were comparable with other preclinical models of EtOH exposure, generally administered via gavage [34,35].

From day E5, pregnant females were fed standard chow (Rat & Mouse Meat-Free Diet, Specialty Feeds). We have previously reported that this model results in equivalent caloric intake and weight gain between treatment and control animals during the treatment period, as well as throughout pregnancy [16]. Ethics approval for all animal experimentation was obtained from the University of Queensland Anatomical Biosciences Animal Ethics Committee (SBMS/467/14/NHMRC) prior to commencement of the study and was conducted in accordance with the *Australian Code for the Care and Use of Animals for Scientific Purposes* (2013, 8th Edition). Reporting of animal experiments conforms to the ARRIVE guidelines [36,37].

Day of birth was designated as postnatal day (PN) 0 and neonatal ovaries were collected from female F1 offspring during early (PN3) and late (PN10) life from 4 litters per treatment/control, as previously described [24]. These time points correspond with peak numbers of primordial follicles and a period of active recruitment to the growing follicle pool in the rat [5]. Only 1–2 females were used from each litter for analysis of gene expression to remove potential litter effects. The remaining 7 litters per treatment/control remained with the dam and were weaned at PN21, with same sex littermates housed together and females used for fertility assessments as described below.

### 4.2. Analysis of Gene Expression for Regulators of Ovarian Reserve in F1 Neonatal Ovaries

Expression of genes for factors with putative roles in regulating primordial follicle growth, survival or activation/recruitment (see Table 2 for more details) were analyzed using qPCR as previously described [24]. Briefly, RNA was extracted using either QIAzol Lysis Reagent (Qiagen, Hilden, Germany) and Glycoblue co-precipitant (Thermo Fisher Scientific, Waltham, MA, USA) for PN3 ovaries (with an additional overnight precipitation at −20 °C) or QIAGEN RNeasy Minikit (Qiagen) according to the manufacturer’s instructions. cDNA synthesis was performed using the iScript RT Supermix (Bio-Rad Laboratories, Hercules, CA, USA) using 200 ng of RNA per reaction. RT reactions were performed on a PCR Express Thermal Cycler (Thermo Fisher Scientific). qPCR reactions were performed on an Applied Biosystems Quantstudio 6 Flex Real-Time PCR System (Thermo Fisher Scientific) using 4 ng cDNA, Taq PCR Master Mix (Qiagen; catalogue #201443) and Taqman Assay-on-Demand primer/probe sets (Thermo Fischer Scientific; see Appendix A for details) per 10 μL reaction. Nine commonly used endogenous control ‘housekeeper’ genes were tested for stable expression across all samples, with only *Hprt* identified as not being regulated by ethanol treatment at both time points (Figure 5). Relative gene expression was determined using the comparative threshold method (ΔΔCT) and normalized to *Hprt.* Fold change was expressed relative to the average of the control group at each age.

### 4.3. Measurement of Estrous Cyclicity and Fertility in F1 Females

As described above, F1 female offspring from 7 litters/group were grown to 3 months of age and estrous cyclicity assessed over 25–27 days via measurement of vaginal impedance using an EC40 estrous cycle monitor (Fine Science Tools), as previously described [24]. Measurements with an impedance >400 kΩ indicate the female is in proestrus [56].

F1 females were mated and breeding performance was assessed across two consecutive litters. F2 pup weights and survival were also measured at PN3, PN10 and at weaning in a subset of litters. Only 1–2 females were used from each litter to remove potential litter effects for all measures. Remaining offspring were used in previously published experiments examining metabolic and behavioral outcomes (see Introduction).

### 4.4. Statistical Analysis

All statistical analyses were conducted using GraphPad Prism Version 7.0 Software (GraphPad Software). Ratios of pro-apoptotic and pro-survival factors (i.e., *Bcl2*:*Bax*; *Bcl2l1*:*Bax*) were calculated, as these are commonly used to examine potential imbalances in cell fate/programmed cell death [57]. Data sets were checked for a normal distribution using the D’Agostino and Pearson test or Shapiro–Wilk test (for small sample size) before conducting parametric or non-parametric analysis as appropriate. Control and PCEtOH groups were compared for each parameter using a Student’s *t*-test (parametric data) or a Mann–Whitney *U*-test (non-parametric data). Data are presented as mean ± SEM, with the exception of the analysis of the endogenous control genes which are presented as box plots with median, inter-quartile range (IQR; 25–75%) and 10–90%-ile shown. Data was considered statistically significant at *p* < 0.05 and considered a trend at 0.05 < *p* < 0.01.

## Figures and Tables

**Figure 1 ijms-25-02471-f001:**
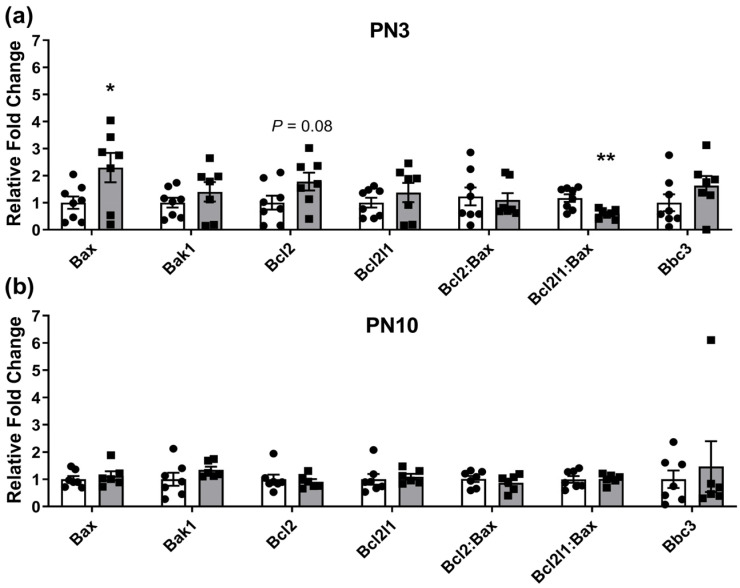
Genes involved in primordial follicle survival in F1 neonatal ovaries. (**a**) PN3 ovaries; (**b**) PN10 ovaries. Closed circles (white bars) are controls; closed squares (grey bars) are periconceptional ethanol exposed (PCEtOH) offspring. Data expressed as mean ± SEM. * *p* < 0.05; ** *p* < 0.01; unpaired *t*-test or Mann–Whitney rank sum test used for each comparison. n = 6–8 across 4 litters/group. PN = post-natal day.

**Figure 2 ijms-25-02471-f002:**
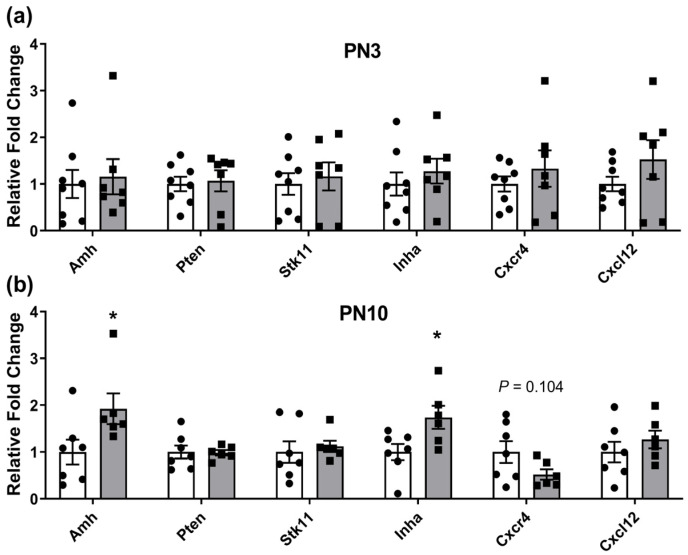
Genes involved in primordial follicle growth and activation/recruitment in F1 neonatal ovaries. (**a**) PN3 ovaries; (**b**) PN10 ovaries. Closed circles (white bars) are controls; closed squares (grey bars) are periconceptional ethanol exposed (PCEtOH) offspring. Data expressed as mean ± SEM. * *p* < 0.05; unpaired *t*-test or Mann–Whitney rank sum test used for each comparison. n = 6–8 across 4 litters/group. PN = post-natal day.

**Figure 3 ijms-25-02471-f003:**
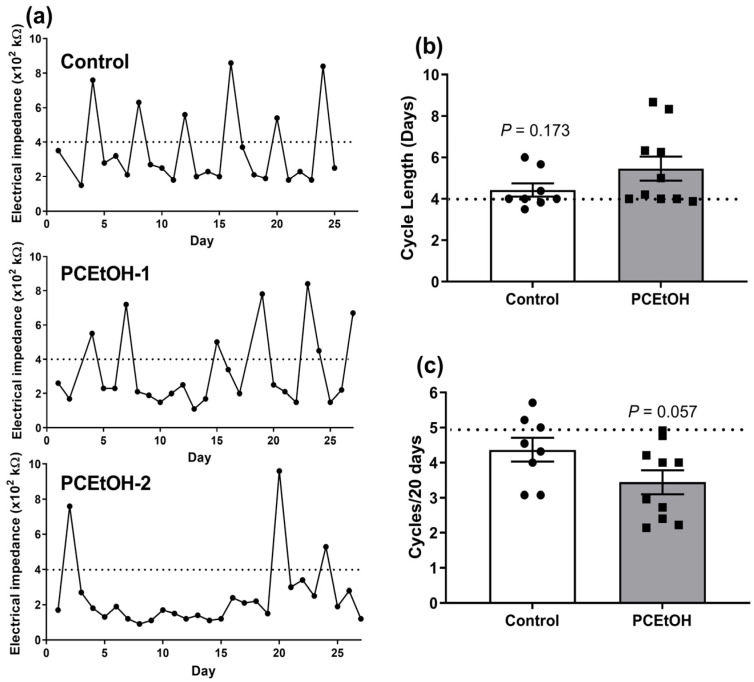
Estrous cyclicity in F1 offspring. (**a**) Estrous cycles were examined in 3-month-old offspring over 25–27 days via measurement of vaginal impedance (>400 kΩ = proestrus, as indicated by the dotted line). The top figure is a representative profile for a control group female, while the bottom two figures are representative of periconceptional ethanol (PCEtOH) group females. (**b**) Mean ± SEM estrous cycle length in days. Dotted line indicates typical 4-day cycle for Sprague Dawley rats. (**c**) Number of cycles over a 20-day period. Dotted line indicates typical 5 cycles per 20-day period in this species. *p* values obtained using a Mann–Whitney rank sum test. n = 8–10 across 7 litters/group.

**Figure 4 ijms-25-02471-f004:**
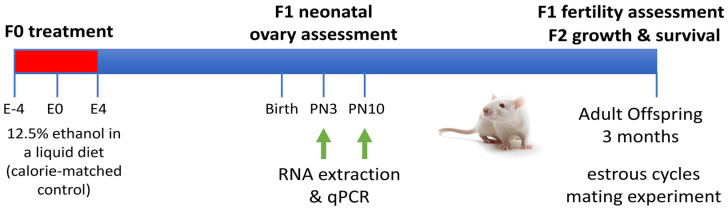
Rat model of periconceptional alcohol exposure. The period of exposure is shown in red. E = embryonic day; PN = post-natal day.

**Figure 5 ijms-25-02471-f005:**
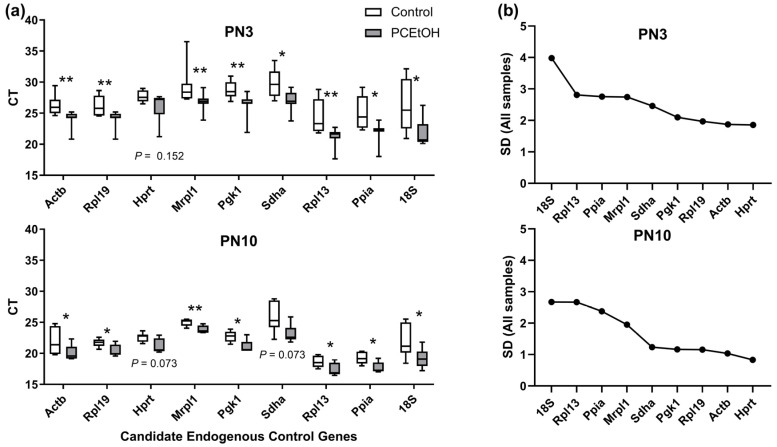
Candidate endogenous control gene expression varied with ethanol exposure. n = 6–8/treatment/age across 4 litters/group. (**a**) Cycle threshold (CT) displayed as median, IQR (25–75% = box) and 10–90% (whiskers). All genes analyzed using Mann–Whitney rank sum tests due to non-normal distribution of most data sets. * *p* < 0.05, ** *p* < 0.01. (**b**) Standard deviation (SD) graphed across all samples for each gene to further illustrate heterogeneity in expression.

**Table 1 ijms-25-02471-t001:** F1 female breeding data and F2 pup weights and survival.

Parameter—Mean (SD)	Control (n = 7) ^1^	PCEtOH (n = 7) ^1^	*p* Value ^2^
F1 age at 1st mating (weeks)	14.71 (1.9)	14.63 (2.1)	0.937
Pups per litter (Litter 1) ^3^	16 (2.4)	16 (2.2)	0.648
Pups per litter (Litter 2) ^3,4^	16 (3.1)	14 (2.4)	0.364
Survival of F2 pups to PN10 (%) ^5^	96.79 (3.2)	89.86 (27.3)	0.837
F2 Pup weight at PN3 (g)—females ^6^	8.19 (1.1)	8.34 (0.9)	0.794
F2 Pup weight at PN3 (g)—males ^6^	8.83 (1.4)	8.85 (1.1)	0.973
F2 Pup weight at weaning (g)—females ^7^	44.36 (3.0)	40.06 (2.9)	0.085
F2 Pup weight at weaning (g)—males ^7^	47.22 (2.3)	44.27 (2.6)	0.140

^1^ One F1 female was randomly chosen from each treated F0 female. ^2^
*p* values were obtained from either an unpaired *t*-test (parametric data) or a Mann–Whitney rank sum test (non-parametric data). ^3^ Pups per litter expressed as whole pups. ^4^ n = 6 per group. ^5^ Across 13 litters/group. ^6^ Across 9 litters/group. ^7^ Across 4 litters/group. F1/F2 = 1st/2nd generation offspring. PCEtOH = periconceptional ethanol. PN = postnatal day.

**Table 2 ijms-25-02471-t002:** Details of genes examined in neonatal ovaries from pregnancies exposed to periconceptional ethanol. Many of these functions have been discovered using knockout mouse models but their precise molecular action and regulation is still under investigation.

Gene Name	Gene Symbol	Function Related to Ovarian Reserve	Reference(s)
**Apoptotic pathway**			
BCL2 associated X, apoptosis regulator	*Bax*	Pro-apoptotic factor	[38,39]
BCL2 antagonist/killer 1	*Bak1*	Pro-apoptotic factor	[40,41]
BCL2 apoptosis regulator	*Bcl2*	Pro-survival factor	[39,42,43]
BCL2 like 1	*Bcl2l1* (*Bclx*)	Pro-survival factor	[44,45]
BCL2 binding component3	*Bbc3* (*Puma*)	Pro-apoptotic via activation of *Bax*/*Bak1* or inhibition of pro-survival BCL2 family members	[46,47]
**Follicle growth and recruitment**			
Anti-mullerian hormone	*Amh*	Maintenance of ovarian reserve by inhibiting recruitment to the growing follicle pool	[48,49,50]
Phosphatase and tensin homolog	*Pten*	Maintenance of quiescent state of primordial follicles via the PI3K/AKT pathway	[51,52]
Serine/threonine kinase 11	*Stk11* (*Lkb1*)	Maintenance of quiescent state of primordial follicles via mTOR	[53]
Inhibin subunit alpha	*Inha*	Indirectly impacts on follicle growth and recruitment via negative feedback of FSH	[54,55]

AKT = protein kinase a; FSH = follicle stimulating hormone; mTOR = mammalian target of rapamycin; PI3K = phosphatidylinositol-3 kinase.

## Data Availability

All underlying data used for figures and tables reported in this study are available from the authors by request.

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
