# Peer review of "The Impacts of Periconceptional Alcohol on Neonatal Ovaries and Subsequent Adult Fertility in the Rat"

_ijms, 2024, doi:10.3390/ijms25052471_

Round 1

Reviewer 1 Report

Comments and Suggestions for Authors

ID: IJMS-2856703

Specific comment below

Introduction

The impact of alcohol on ovarian cell reduction or growth should be expanded as a fundamental understanding of whether alcohol impacts cell growth or apoptosis. This must be highlighted because the gene expression analysis revealed that the majority of the cells were in apoptotic state. Make it clear in the introduction. The study's goal is to investigate the effect of alcohol on cell regulation, particularly in the ovaries.

Material methods

The alcohol dosage is 12.5% v/v. Add a reference or a reason for using this.

The non-significant result could be attributed to the low concentration of alcohol.

Author Response

Thank you for reviewing our manuscript and the opportunity to provide further clarification. We have addressed your comment on our introduction by re-wording the aims of the study at lines 58-60. We have addressed your comment on the justification for the dose in the methods at lines 209-216.

Reviewer 2 Report

Comments and Suggestions for Authors The manuscript describes the periconceptional maternal alcohol exposure on the genes that condition ovarian reserve in the F1 generation, as well as the length of the estrus cycle of the F1 generation and the weight and survival rate of the 2nd generation. While the manuscript is interesting, the major weakness of this study is that it only focuses on gene expression, which may only indicate potentail to protein changes. There are no histologic or protein analyzes. The manuscript should have included a clear explanation of why the authors chose 12.5% v/v EtOH and how this dose correlates with "consumption of low to moderate amounts of alcohol around the periconceptional period and before pregnancy recognition” (lines 45-47). Are there any considerations for this amount of EtOH in studies with other animal species? Since the model was ad libitum, are the authors certain that all females consumed the approximate amount of EtOH? Why did the authors focus on these specific genes in PN3 and PN10, while according to Picute et al. 2014, "During the neonatal stage (postnatal day [PND] 0–7), ovarian follicle development is independent of pituitary gonadotropins (luteinizing hormone [LH] or follicle-stimulating hormone [FSH]), and follicles remain preantral. Antral development of “atypical” follicles occurs in the early infantile period (PND 8–14) when the ovary becomes responsive to pituitary gonadotropins". The aim of the study should therefore be formulated more precisely. And why did not the authors also analyze the other genes associated with ovarian cell activity, e.g. FSHR and LHR?

Author Response

Thank you for reviewing our manuscript and the opportunity to provide further clarification. The major points are addressed below:

  • We have acknowledged that we are only focussing on gene expression changes in the abstract by adding at line 16/17 “…genes with protein products involved in…”. We have also made mention of this in the discussion at lines 125-126 and lines 152-153.
  • We have previously reported the explanation for our choice of dose (i.e. 12.5% v/v EtOH) in our first paper using this model (i.e. Ref #31, Gardebjer et al., 2014). However, given another reviewer also commented on this, we have added some additional information with regards to our choice of dose at lines 209-216. When comparing across species, it is best to consider the blood alcohol concentration (BAC) produced rather than the dose per se, as metabolism of alcohol varies across species. Therefore, we mention the range of BACs produced in this model and how this equates to human BACs. This is how we determined that the level was “low to moderate”.
  • With respect to the ad libitum consumption of the diet, we have provided some evidence from our previous publication in the methods at lines 205-208 to suggest that all animals consumed a similar amount of diet over the treatment period.
  • With regards to our choice of genes, we agree with your statement that during the neonatal period, follicle development/growth is independent of LH and FSH and therefore maintenance or activation of primordial follicles occurs via intrinsic factors produced within the ovary, specifically by granulosa cells of preantral follicles. While there is some evidence of ‘atypical’ antral follicle development later in the neonatal period, as we have also shown in our previous publication at the PN10 timepoint, our focus was on genes previously reported to be important in maintaining the ovarian reserve (i.e. that produce factors that may act on primordial follicles), rather than genes that may be involved later in folliculogenesis (i.e. growth from early antral to antral when FSH and LH are important). Therefore, we thank you for the suggestion to re-word our aims to more precisely describe the particular types of genes examined and that we are focussed on factors regulating maintenance of the ovarian reserve, rather than later stages of follicular growth. This has been done at lines 58-60.

Reviewer 3 Report

Comments and Suggestions for Authors

I would like to ask the Authors to answer/correct some issues listed below.

Lines 120-130 and the whole discussion section: I would like the Authors to address the hypothesis assuming that the PCEtOH exposure could limit the oocyte reserve in F1 and F2 generations. You showed that let’s call it the “treated group” shows a longer cycle (although it was statistically not significant) and fewer cycles during the same period compared to the control group. Could it be an effect of lower ovarian reserve in these animals? Did the Authors check further litters of F1 and F2 generations? How long were they able to reproduce? Could alcohol exposure create a state similar to “earlier menopause”? Could it have similar implications in humans? And, finally, did the Authors check the “treated group” reproductive performance from a long-term perspective?

Minor concerns:

Line 16: it should be ”…genes whose protein products are involved…”

Line 19: You should not use shortcuts without explaining them first “PN3”

Line 20: “ gene expression changes…”

Author Response

Thank you for reviewing our manuscript and your helpful comments. To address your concerns re potential impacts on the ovarian reserve and long-term fertility we have added a paragraph in the discussion at lines 152-170. Given we did not examine the reproductive capacity of the F2 generation of females, only their initial growth via measurement of weights at birth and weaning, we are hesitant to speculate on potential impacts beyond the F1 generation.

We have also addressed your minor concerns with lines 16, 19 and 20 (now line 21) as suggested, with a slight edit to your suggestion at line 16-17.

Reviewer 4 Report

Comments and Suggestions for Authors

This is well conducted study. The research team for sure is experienced in area of published experiment. Moreover, important and valuable is observation from submitted study. Results may be extrapolated on human studies, i.e. alcohol impact and consumption in early pregnancy which is often a reason of high level of woman fears. References are up to date and adequate. I would strongly recommend and encourage further experiments in this field. 

Author Response

We thank you for your positive comments.

Round 2

Reviewer 2 Report

Comments and Suggestions for Authors

The authors increased the quality of the manuscript and responded to all my queries.